

# Mechanisms of oat (*Avena sativa* L.) acclimation to phosphate deficiency

Ewa Żebrowska[1,2,*], Marta Milewska[1] and Iwona Ciereszko[1,*]

[1] Department of Plant Physiology, Institute of Biology, University of Bialystok, Bialystok, Poland
[2] Department of Physiology, Medical University of Bialystok, Bialystok, Poland
[*] These authors contributed equally to this work.

## ABSTRACT

**Background**. Deficiency of available forms of phosphorus is common in most soils and causes reduction of crop plants growth and yield. Recently, model plants responses to phosphate (Pi) deficiency have been intensively studied. However, acclimation mechanisms of cereals like oat (*Avena sativa* L.), to low Pi stress remains not fully understood. Oat plants have been usually cultured on poor soils, with a low nutrient content, but their responses to such conditions are not well known, therefore the main goal of the study was to investigate the mechanisms that enable oat plants to grow under low Pi conditions.

**Methods**. Four oat cultivars (*A. sativa*, cv. Arab, Krezus, Rajtar and Szakal) were grown for three weeks in a nutrient media with various P sources: inorganic— $KH_2PO_4$ (control), organic—phytate (PA) and with no phosphate ($-P$). The effects of Pi deficiency on the level of P, oat growth parameters, intensity of photosynthesis, plant productivity, root exudation ability, localization, activity and isoforms of acid phosphatases, enzymes involved in Pi mobilization, were estimated. In addition, the effect of mycorrhization on plant growth was also observed.

**Results**. All studied oat cultivars grown on Pi-deficient media had significantly decreased Pi content in the tissues. Pi deficiency caused inhibition of shoot growth, but generally it did not affect root elongation; root diameter was decreased, root/shoot ratios increased, whereas PA plants showed a similar growth to control. Photosynthesis rate and productivity parameters decreased under low Pi nutrition, however, sugar content generally increased. Studied oat cultivars did not respond to low Pi *via* increased exudation of carboxylates from the roots, as pH changes in the growth media were not observed. Pi starvation significantly increased the activity of extracellular and intracellular acid phosphatases (APases) in comparison to the control plants. Three major APase isoforms were detected in oat tissues and the isoform pattern was similar in all studied conditions, usually with a higher level of one of the isoforms under Pi starvation. Generally no significant effects of mycorrhizal colonization on growth of oat cultivars were observed.

**Discussion**. We postulated that acid phosphatases played the most important role in oat cultivars acclimation to Pi deficiency, especially extracellular enzymes involved in Pi acquisition from soil organic P esters. These APases are mainly located in the epidermis of young roots, and may be released to the rhizosphere. On the other hand, intracellular APases could be involved in fast Pi remobilization from internal sources. Our study showed that oat, in contrast to other plants, can use phytates as the sole source of P. The studied oat cultivars demonstrated similar acclimation mechanisms to

Corresponding author
Iwona Ciereszko, icier@uwb.edu.pl

Pi deficiency, however, depending on stress level, they can use different pools of acid phosphatases.

# INTRODUCTION

Plants require a wide range of mineral nutrients for normal growth and development. Phosphorus (P) is one of the least accessible macronutrients. It plays key roles in many plant processes, such as organic compound synthesis—photosynthesis and respiration or regulation of enzyme activities and gene expression (*Vance, Uhde-Stone & Allan, 2003*; *Rychter & Rao, 2005*; *Hammond & White, 2008*; *Amtmann & Armengaud, 2009*). Plants absorb only inorganic phosphate, Pi ($HPO_4^{2-}$ and $H_2PO_4^-$) therefore, although the total amount of P in the soil may be high, its availability for plants is low (*Schachtman, Reid & Ayling, 1998*; *Vance, Uhde-Stone & Allan, 2003*; *Raghothama & Karthikeyan, 2005*). In acid soils, phosphorus forms poorly soluble complexes with iron (Fe), and aluminium (Al), while in neutral to alkaline soils it combines with calcium (Ca) (*Holford, 1997*; *Smith et al., 2003*). It is assumed that organic P (mainly phytate and its derivatives), representing even up to 80% of total P in soil, is also unavailable for plants (*Schachtman, Reid & Ayling, 1998*; *Gerke, 2015*). Phosphate deficiency is thus one of the main limiting factors for plant productivity and crop yield. Modern agriculture strongly depends on P-fertilizer application, however, plants can absorb only up to 25% of Pi from mineral fertilizers (*Hermans et al., 2006*; *Lynch, 2011*; *Shenoy & Kalagudi, 2005*). Furthermore, the resources of P for fertilizer production (mined rock phosphates) are non-renewable and the longevity of phosphate reserves is limited (up to several hundred years at current production rates) (*Gerke, 2015*; *Cordell, Drangert & White, 2009*). Moreover, fertilizer prices are rising mainly due to the expensive extraction of low quality phosphate rock and dependency of most countries on fertilizer import (the vast majority of global reserves is held only by a few countries). Currently, most scientists agree that there is a strong need for increased recycling and efficient use of phosphorus to sustain or extend food production in the context of a growing human population (*Cordell, Drangert & White, 2009*; *Lynch, 2011*; *Scholz, Roy & Hellums, 2014*; *Faucon et al., 2015*). Therefore, the improvement of Pi acquisition and utilization efficiency by crop plants is now fundamental for solving problems of Pi deficiency.

Plants have evolved various morphological, physiological and biochemical adaptive responses to overcome phosphate deficiency that include increased Pi uptake from the soil and/or more efficient Pi use in the tissues (*Smith et al., 2003*; *Raghothama & Karthikeyan, 2005*; *Hammond & White, 2008*; *Richardson et al., 2009*; *Tran, Hurley & Plaxton, 2010*; *Faucon et al., 2015*). Efficient use of acquired Pi in plant tissues under Pi starvation is possible through a variety of physiological and metabolic adaptations, such as alternative metabolic photosynthetic pathways, glycolysis and mitochondrial electron transport or

enhanced photorespiration and carbon metabolism (*Maleszewski et al., 2004*; *Ciereszko & Kleczkowski, 2005*; *Rychter & Rao, 2005*; *Amtmann & Armengaud, 2009*; *Plaxton & Tran, 2011*). Plants grown under Pi deficiency can also allocate a greater proportion of assimilates to root growth at the expense of shoot growth (*Ciereszko, Miłosek & Rychter, 1999*; *Hermans et al., 2006*; *Hammond & White, 2008*). It was demonstrated that even small changes in root morphology could be important for better exploration of soil and influences Pi uptake (*Wissuwa, 2003*; *Lynch, 2011*; *Péret et al., 2014*; *Stetter, Martin Benz & Ludewig, 2017*). Changes in root morphology/architecture under Pi deficiency mainly include: an increase or decrease of root length, promotion of lateral root growth, enhancement of root hair development and proteoid root (clusters of lateral roots) formation (*Neumann et al., 2000*; *Williamson et al., 2001*; *Niu et al., 2013*). Proteoid roots not only increase root surface area, but also secrete many compounds, such as organic acids and acid phosphatases that increase Pi availability (*Lambers et al., 2012*; *Lambers et al., 2015*). In response to Pi starvation, plants produce a shallow branched root system easily exploring large areas of the upper layer of soil in search of Pi-rich patches. Recently, several studies have addressed the function played by transcription factors, such as MYB62, WRKY75, PHR1 and ZAT6 or miRNAs and phytohormones in root architecture modification in response to low Pi nutrition (*Tran, Hurley & Plaxton, 2010*; *Niu et al., 2013*; *Péret et al., 2014*; *Zhang, Liao & Lucas, 2014*; *Baker et al., 2015*; *Stetter, Martin Benz & Ludewig, 2017*). Recent studies have also shown that root architecture changes are controlled by local signals of Pi availability in the soil rather than internal Pi content in tissues (*Thibaud et al., 2010*; *Hoehenwarter et al., 2016*). Root surface area and the ability to uptake Pi from the soil may be also increased by mycorrhizal hyphae. It is estimated that more than 80% of the land plants form arbuscular mycorrhizal symbioses (AM) with soil fungi (mainly Glomeromycota) (*Smith et al., 2003*; *Karandashov & Bucher, 2005*; *Smith & Smith, 2011*). AM fungi stimulate host plant growth, especially by enhancing Pi uptake in order to receive organic compounds. Pi uptake of colonized roots may even be several times higher than in non-infected roots (*Raghothama & Karthikeyan, 2005*; *Shu, Wang & Xia, 2014*; *Baker et al., 2015*). Two pathways of Pi acquisition are known: *via* root epidermal cells and root hairs, and *via* AM fungi that deliver Pi directly to the root cortex and may represent 70% of the total acquired phosphate (*Smith & Smith, 2011*; *Yang et al., 2012*). Efficient Pi uptake requires active Pi transporters, both of fungal and plant origin. It was reported that numerous plant Pi transporter genes could be induced by mycorrhizal fungi, however, their expression varied depending on AM fungi species (*Ceasar et al., 2014*; *Baker et al., 2015*).

Plant roots can secrete organic acids, protons and phenolics that increase the availability of inorganic, poorly soluble forms of P, such as Ca, Fe, and Al phosphates. Malic and citric acids are predominant organic acids found in root exudates of Pi-deficient plants (*Vance, Uhde-Stone & Allan, 2003*; *Richardson et al., 2009*; *Wang et al., 2015*; *Wang et al., 2017*). The enhanced organic acid secretion is determined mainly by increased activities of phosphoenolpyruvate carboxylase, malate dehydrogenase and citrate synthase (*Plaxton & Tran, 2011*; *Baker et al., 2015*). Another mechanism of plant response to Pi deficiency in the environment is increased production and secretion of enzymes that facilitate the hydrolysis of organic forms of phosphorus, particularly extracellular acid phosphatases

(APases) (*Duff, Sarath & Plaxton, 1994*; *Tran, Hurley & Plaxton, 2010*). Acid phosphatases (orthophosphoric monoester phosphohydrolases, EC 3.1.3.2) hydrolyse different forms of organic P, usually in a non-specific manner. APase activity has been shown in cells of various organisms, like bacteria, yeast, and plants (*Duff, Sarath & Plaxton, 1994*; *Żebrowska & Ciereszko, 2009*; *Tran, Hurley & Plaxton, 2010*). APases are involved in the uptake, allocation and recycling of Pi, processes which are crucial for cellular metabolism and bioenergetics. Expression of APase genes is affected by different environmental factors, including Pi deficit; enhanced APase synthesis and activity was also observed under salinity and water-deficit stress (*Tran, Hurley & Plaxton, 2010*). Secretion of APases into the rhizosphere is a typical response to Pi-starvation and has been documented in various models and crop plants, including *Arabidopsis*, barley, lupine, oat, rape, rice, soybean, tomato and wheat (*George et al., 2008*; *Zhang et al., 2010*; *Ciereszko, Szczygła & Żebrowska, 2011*; *Ciereszko, Żebrowska & Ruminowicz, 2011*; *Żebrowska, Bujnowska & Ciereszko, 2012*; *Del Vecchio et al., 2014*; *Tian & Liao, 2015*; *Lu et al., 2016*). The high secretion of acid phosphatases was observed in white lupine roots, especially in proteoid regions (*Wasaki et al., 2009*; *Tang et al., 2013*). What is more, extracellular APase activity was shown to increase during proteoid root development, whereas internal APase activity was relatively constant (*Tang et al., 2013*). Purple acid phosphatases are the most investigated enzymes among APases (*Olczak, Morawiecka & Watorek, 2003*; *Tran, Hurley & Plaxton, 2010*), e.g., three of them (AtPAP10, AtPAP12 and AtPAP26) can be secreted under Pi deficiency from *A. thaliana* roots (*Del Vecchio et al., 2014*; *Tian & Liao, 2015*). Purple APases may also participate in cell wall biosynthesis, carbon metabolism or biotic stress tolerance (*Tran, Hurley & Plaxton, 2010*; *Tian & Liao, 2015*). Under low Pi conditions, the activity of internal APases (pivotal for Pi mobilizing from P-rich cell organelles) is also increased (*Tran, Hurley & Plaxton, 2010*; *Zhang et al., 2010*; *Ciereszko, Szczygła & Żebrowska, 2011*; *Ciereszko, Żebrowska & Ruminowicz, 2011*; *Tian & Liao, 2015*). Pi deficiency can also enhance the release of specific enzymes, such as phytases (*myo*-inositol hexakisphosphate phosphohydrolase, EC 3.1.3.8, 3.1.3.26), that catalyse the hydrolysis of phytates, or nucleases and apyrases that hydrolyse nucleic acids and extracellular ATP in the soil (*Vance, Uhde-Stone & Allan, 2003*; *Richardson et al., 2009*; *Gerke, 2015*).

Hundreds of genes associated with plant responses to Pi deficiency have been identified in the model plant *A. thaliana* and certain crop plants, mainly rice (*Misson et al., 2005*; *Thibaud et al., 2010*; *Li et al., 2012*; *Péret et al., 2014*; *Zhang, Liao & Lucas, 2014*). Recently, Pi signalling networks and signalling molecules have been intensively studied (*Thibaud et al., 2010*; *Zhang, Liao & Lucas, 2014*; *Baker et al., 2015*; *Hoehenwarter et al., 2016*). Quantitative trait loci analyses show that Pi acquisition traits are complex and regulated by multiple genetic loci and so far have not been often used by breeders (*Wissuwa, 2003*; *Lynch, 2011*; *Niu et al., 2013*). Recently, with development of transgenic methods, the number of genes improving P efficiency that have been successfully introduced into crop species has been constantly rising in laboratory conditions (*Tran, Hurley & Plaxton, 2010*; *Niu et al., 2013*; *Zhang, Liao & Lucas, 2014*). However, even interesting transgenic lines of cereals are not commercially available, not only because of technical problems but also due to the public opposition to genetically modified food.

Mechanisms of plant acclimation to phosphate deficiency are well documented for model plants, however, there are not many studies on acclimation of cereal plants, such as oat, to Pi deficiency. Oat (*Avena sativa* L.) is an important crop plant in agriculture, human and animal nutrition as well as cosmetic and pharmaceutical industry (*Butt et al., 2008*). Oat plants have been always grown on worse, less productive agricultural lands and poor soils, with a low Pi concentration, but their acclimation mechanisms to such conditions are still not fully understood. Therefore, the main goal of our study was to provide a comprehensive analysis of these mechanisms and to assess the differences between oat cultivars. We investigated a wide array of physiological responses of four oat cultivars to early and intermediate stress of Pi deficiency during a period critical to tiller formation and further plant productivity. In particular, we focused on the activity (and localization) of acid phosphatases to evaluate their role in oat response to Pi deficiency. The present study also investigated the effects of phytate as the sole organic P source in the nutrient medium on the growth of oat plants. With rising P-fertilizer prices, selecting crop cultivars with improved nutrient acquisition and P-use efficiency is an important component of an integrated strategy for solving the problem of phosphate deficiency.

## MATERIALS & METHODS

### Plant material

Four oat (*Avena sativa* L.) commercial cultivars (recommended for farmers): Arab (registered in 2004), Krezus (2005) Rajtar (2004) and Szakal (2000) were selected, among several other oat genotypes, in the preliminary experiments including APase secretion, and used for the further studies. Oat seedlings, after 6–7 days of germination (Petri dishes, in a growth chamber), were grown for 1–3 weeks in nutrient media with contrasting phosphorus source: inorganic—$KH_2PO_4$ (control, +P), organic—phytic acid (PA) and with no phosphate (−P) as described previously (*Ciereszko, Szczygła & Żebrowska, 2011*; *Ciereszko, Zebrowska & Ruminowicz, 2011*). +P medium contained: $Ca(NO_3)_2$ (4.4 mM), $MgSO_4$ (2.7 mM), $KNO_3$ (1.5 mM), $KH_2PO_4$ (1 mM), Fe-EDTA (76 μM), $H_3BO_3$ (43 μM), $MnCl_2$ (9 μM), $CuSO_4$ (0.3 μM), $ZnSO_4$ (0.8 μM), $H_2MoO_4$ (0.1 μM); PA medium contained phytic acid (0.1 mM) (instead $KH_2PO_4$; concentration chosen after preliminary studies) and −P medium—KCl (2 mM) (instead $KH_2PO_4$). Oat cultivars were cultured in separate plastic containers (about 15 seedlings per 5 l of nutrient medium); growth media were adjusted to pH 5.2 (adding drops of 1N NaOH), continuously aerated and replaced every 4–5 days. Oat plants were cultured under controlled conditions—in a growth chamber with a light period of 16 h (8 h dark), PAR —150 μmol m$^{-2}$ s$^{-1}$, temperature —23/19 °C (day/night) and relative humidity around 60%. The plant material was collected 3–4 h after the beginning of the light period. Growth analyses were performed directly, and the following growth parameters were calculated: leaf area ratio (LAR), relative growth rate (RGR), stem weight ratio (SWR) and unit leaf rate (ULR), according to the method of *Miranda, Fischer & Ulrichs (2011)*. Root exudation was determined by measuring the pH of nutrient media and rhizosphere acidification on agar sheets with bromocresol purple, as described by *Neumann et al. (2000)* (Fig. S3). Oat plants were harvested for analyses after 1, 2 and 3 weeks of growth in various nutrient media.

## Phosphorus content measurements

Inorganic phosphate (Pi) content in the leaves and roots of oat cultivars under the studied conditions was determined using a phosphomolybdate colorimetric assay, as described before (*Ciereszko, Szczygła & Żebrowska, 2011*; *Ciereszko, Zebrowska & Ruminowicz, 2011*). Aliquots of 0.5 g of roots/leaves were frozen in liquid nitrogen and homogenized in 10% (v/v) TCA at 4 °C, diluted with 5% TCA, incubated for 30 min and then centrifuged (10 min at 10,000× g). The Ames reagent was added to the tissue extract, the samples were incubated for 20 min at 25 °C and absorbance was read at 710 nm (Beckman DU 730). Total phosphorus (Pt, Fig. S1) was determined in tissue samples (0.05 g dry weight) after mineralization with concentrated $H_2SO_4$ and $HNO_3$; Pi content was determined as described above.

## Photosynthesis rate estimations

Measurements were taken using a LI-COR $CO_2$/$H_2O$ analyser (LI-6262, LI-COR Inc., Lincoln, NE, USA) under conditions similar to seedling cultures: photosynthetic photon flux density 200 $\mu$mol (photon) m$^{-2}$ s$^{-1}$, $O_2$ and $CO_2$ at atmospheric concentrations, temperature of 25 °C, as described previously (*Maleszewski et al., 2004*). Chlorophylls and carotenoids were extracted from the leaves with 100% methanol at 70 °C and determined spectrophotometrically, according to the method of *Wellburn (1994)*.

## Carbohydrate content determination

For soluble sugar extraction, shoot and root samples (0.5 g) were frozen in liquid nitrogen, then ground in 80% ethanol and incubated for 20 min at 70 °C, subsequently centrifuged (10 min, 10,000× g) and washed three times with 80% ethanol. The precipitate was used for starch measurements, whereas the supernatant was evaporated and the residue was washed/diluted with distilled water and chloroform (1:1) and used for determination of soluble sugars as described before (*Ciereszko, Janonis & Kociakowska, 2002*).

For starch determination, the precipitates were dried at 60 °C for two hours, then 2 ml 0.2 M KOH was added and boiled in a water bath for 1.5 h. After cooling, 0.2 ml 4 M $CH_3COOH$ and 1 U of amyloglucosidase in 0.1 M acetate buffer (pH 4.6) was added. Samples were incubated at 37 °C for 48 h then heated 10 min in a boiling water bath. The amount of glucose (hydrolysed by amyloglucosidase from starch) was determined as described before (*Ciereszko & Barbachowska, 2000*).

## Extracellular acid phosphatase activity

Intact root systems of oat cultivars were washed (distilled water), dried and incubated at about 25 °C with a substrate mixture (6 mM *p*-nitrophenyl phosphate (*p*NPP) and 1 mM dithiothreitol in 50 mM Na-acetate buffer, pH 5.0), (30 or 50 ml of medium, depending on root size), according to the method described by *Żebrowska, Bujnowska & Ciereszko (2012)*. After incubation, 0.2 ml of 4 M NaOH was added (to stop the reaction) to 0.2 ml of the reaction medium; the absorbance was read at 410 nm (Cecil CE 2501). The measurements were performed after 15 min of incubation ($\mu$mol *p*NP min$^{-1}$g$^{-1}$FW). Kinetic assay, the Lineweaver-Burk plot and the $K_m$ and $V_{max}$ were estimated for extracellular APase activity

in the conditions referred above; the concentration of $p$NPP ranged from 0 to 10 mM (Fig. S4).

## Intracellular APase activity assays

For enzyme extraction, 0.5 g of shoots and roots were ground in liquid nitrogen, the extraction buffer (50 mM Na-acetate, pH 5.0 with 1 mM DTT) was added, then extracts were centrifuged at 12,000× g for 10 min at 4 °C (*Ciereszko, Janonis & Kociakowska, 2002*; *Żebrowska, Bujnowska & Ciereszko, 2012*). The reaction was terminated with 4 M NaOH after 5–60 min of incubation with 6 mM $p$ NPP (in 100 mM Na-acetate buffer, pH 5.0) at 37 °C and the amount of $p$-nitrophenol was measured as described above. The measurements were taken after 15 min of incubation ($\mu$mol $p$NP min$^{-1}$g$^{-1}$FW).

## Tissue localization of acid phosphatases

For tissue localization of APases, hand-made cross sections of oat roots (at the maturation zone) were rinsed in Na-acetate buffer (0.1 mM, pH 5.0) and incubated in a substrate mixture (0.2% Fast Blue B, 0.2% 1-naphthyl phosphate,100 mM Na-acetate buffer, pH 5.0). After incubation (20 min), tissues were washed (distilled water) and photographed under a light microscope (Olympus BX41). A dark red-brown colour indicated acid APase activity in the root tissues (*Żebrowska, Bujnowska & Ciereszko, 2012*).

## Analysis of acid phosphatase isoforms

Root and shoot tissues of four oat cultivars (1 g) were ground in a mortar chilled with liquid $N_2$, then 4 ml of extraction buffer was added (100 mM Na-acetate, pH 5.0, 2 mM EDTA, 20 mM $CaCl_2$, 5 mM DTT and 60 mg PVPP), the solution was gently mixed at 4 °C for 60 min and subsequently centrifuged at 10,000 rpm, then pellet was discarded. Equal protein amounts (10 $\mu$g for individual shoots per lane and 6 $\mu$g for roots) were loaded onto a discontinuous native PAGE (5% (w/v) stacking gel, 10% (w/v) resolving gel). The native gels were run at 4 °C using a mini-gel system (SE 260; Hoefer, Holliston, MA, USA; Amersham, Little Chalfont, UK) and washed in 0.1 mM Na-acetate buffer. Approximate masses of APase isoforms were determined using Full Range Rainbow Molecular Weight Markers (Amersham). The fluorescence of methylumbelliferone liberated by phosphatase activity was visualized under UV light (Gel Doc 2000, ver.4.1; Bio-Rad, Hercules, CA, USA), as described before (*Żebrowska, Bujnowska & Ciereszko, 2012*). Protein concentration in shoot and root extracts was determined according to the *Bradford (1976)* method, at 595 nm (CE 2501; Cecil, Cambridge, UK), with BSA as the standard.

## Effect of mycorrhizal colonization on plant growth and Pi content

Seedlings of two oat cultivars (Arab and Krezus), treated or not treated (control) with mycorrhizal fungus *Glomus intraradices* (M), were cultured on a sterile sand mixture. For mycorrhizal infection, root fragments of *Pelargonium* sp. and *Denebola* sp. infected with the fungus *G. intraradices* were mixed with sand (1:10) (*Nowak, 2009*). Plants were watered with nutrient media with different P source: inorganic —$KH_2PO_4$ (control), organic —phytic acid (PA) and without Pi (−P) (as described in the Plant Material section). Mycorrhizal colonization was determined after 4 weeks of culture in a growth chamber

(controlled growth conditions as described before) by staining the roots with trypan blue, as described by *Phillips & Hayman (1970)*. Growth parameters of oat plants and inorganic phosphate content in leaves and roots under all conditions studied was measured using methods described above.

## Statistical analysis

All experiments were performed in 3–5 independent series, at different times, and all assays were carried out at least in three replicates. Standard deviation (SD) was calculated. The data were analyzed by one-way analysis of variance (ANOVA), in addition Duncan's multiple-range test was carried out (Statistica 6, StatSoft, Palo Alto, CA, USA). The significance level in comparisons was $p < 0.05$.

## RESULTS

Oat cultivars (*Avena sativa* L. cv. Arab, Krezus, Rajtar and Szakal) cultured for three weeks on −P nutrient medium showed significantly reduced Pi content and altered growth characteristics. Pi content was severely decreased already after one week of culture on −P medium and after three weeks it was decreased down to 4% of control (+P) in the shoots of all tested cultivars (Figs. 1A–1C) as well as the roots of cv. Arab and Rajtar, whereas in cv. Krezus and Szakal roots it was 8% of control (Figs. 1D–1F). However, the plants grown on phytic acid (PA), as the sole source of phosphorus showed similar Pi content to control plants, especially in shoots (Figs. 1A–1C). The reduction in Pi content in the roots of PA plants occurred after 2–3 weeks of cv. Arab and Krezus culture and was about 46% of control (Figs. 1D–1F). The decrease in Pi content in −P plant tissues was accompanied by a total phosphorus decrease (see Fig. S1). However, Pt content in plants grown on phytic acid was less affected, especially in roots where the Pt level was even higher (cv. Krezus roots, 3 weeks culture). The differences in phosphorus content observed between oat cultivars under study were not derived from initial differences in seed P content (see Fig. S1). Plant growth was strongly affected by phosphate deficiency in plant tissues after 2–3 weeks of culture (Fig. 2, Table 1). Shoot fresh weight of −P plants of all the studied cultivars was decreased down to 40% of control after 2 weeks (Table S1) and after 3 weeks it was reduced to 24% for cv. Arab and 15% for other oat cultivars (Table 1). Root fresh weight after two weeks of −P culture was lowered only in cv. Krezus and Szakal–down to 50–60% of control (see Table S1). After 3 weeks, the lowest fresh mass was observed in cv. Szakal roots (26% of control) (Table 1). The decrease in root mass was not accompanied by the reduction in root length. After 2–3 weeks of growth, root length was approximately 1.3 times higher than in control plants (see Fig. S2). −P plants had a higher root length/shoot height ratio after 2–3 weeks of growth, and it was the highest during the last week of culture (two times higher than in control). Fresh weight and the root to shoot ratios of PA plants were similar to control plants during three-week-culture of oat cultivars studied (Table 1, Table S1).

Growth and productivity parameters of plants cultured on different nutrient media were calculated and were similar after two (Table S1) and three (Table 1) weeks of plant culture. Leaf area ratio (LAR) was reduced only for Arab and Krezus cultivars and (86% and 65%

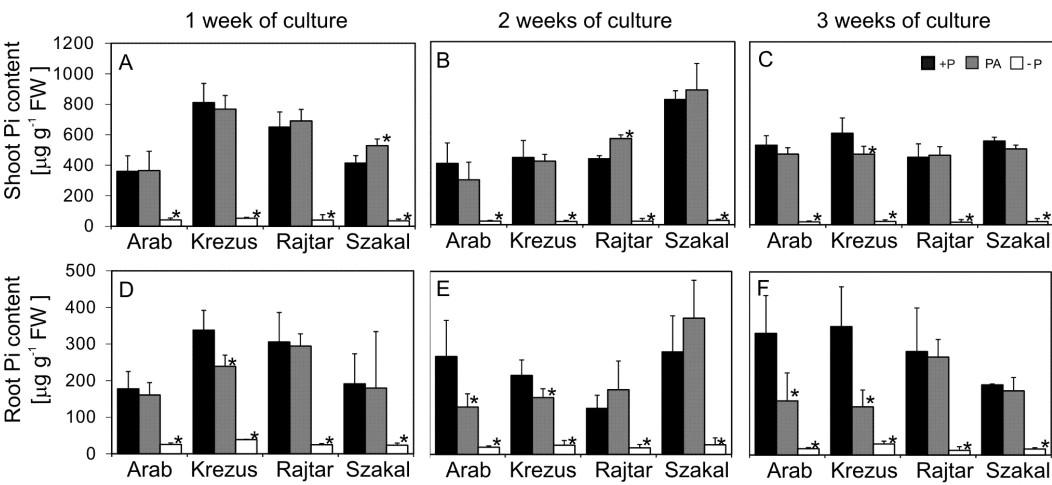

**Figure 1** **Inorganic phosphate (Pi) content in shoots (A–C) and roots (D–F).** Oat cultivars (*Avena sativa L.*, Arab, Krezus, Rajtar and Szakal) grown for 1, 2 and 3 weeks on a complete nutrient medium (+P), medium with phytic acid (PA) and without phosphate (−P) (means ± SD). *Differences statistically significant at 0.05.

of control, respectively) after 3 weeks of growth on −P medium (Table 1). Phosphate deficiency decreased the shoot weight ratio (SWR) of all the studied oat cultivars; after 3 weeks of culture, SWR in −P plants was about 65% for Arab and Krezus, 71% for Rajtar and 78% for Szakal cultivars in comparison to control (Table 1). SWR in PA plants was similar to control during the culture (a decrease to 90% of control was observed only in cv. Rajtar after 3 weeks). Unit leaf rate (ULR) and relative growth rate (RGR) were decreased (down to 32% of control) only for cv. Krezus after 3 weeks of growth on −P medium. Surprisingly, both of these parameters were 1.4 times higher in cv. Arab cultured under phosphate deficiency. PA plants showed similar ULR and RGR to control plants (Table 1).

Phosphate deficiency significantly decreased root diameter up to 70% of control for cultivar Arab, 60% for Krezus and Rajtar and 50% for Szakal after 3 weeks of culture (Table 1). −P plants also had lower root area (except cv. Arab) down to 80% of control for cv. Rajtar, 74% for Krezus and 57% for Szakal. Plants cultured on phytic acid had similar root diameter and surface compared to +P plants (Table 1).

Phosphorus deficiency did not markedly affect net the $CO_2$ assimilation rate ($P_N$) during 1–2 weeks of −P culture, whereas a significant decrease (down to 50–60% of control) was observed for all oat cultivars after 3 weeks of culture (Figs. 3A–3C). These changes were not observed in PA plants. Chlorophylls and carotenoids content was generally not significantly affected by Pi deficiency (see Table S2). A decrease in $P_N$ intensity after 3 weeks of culture was accompanied by soluble sugar accumulation in shoots of all the oat cultivars studied (the highest for cv. Rajtar –1.9 times higher in comparison to +P) (Figs. 3D–3E). Soluble sugar content in the roots of −P plants was similar to control. Phosphate deficiency led to starch accumulation both in shoots (up to 1.8 times higher for cv. Krezus) and roots (1.3 times higher for all cultivars) of −P plants (Figs. 3F–3G). Sugar content in PA plants was

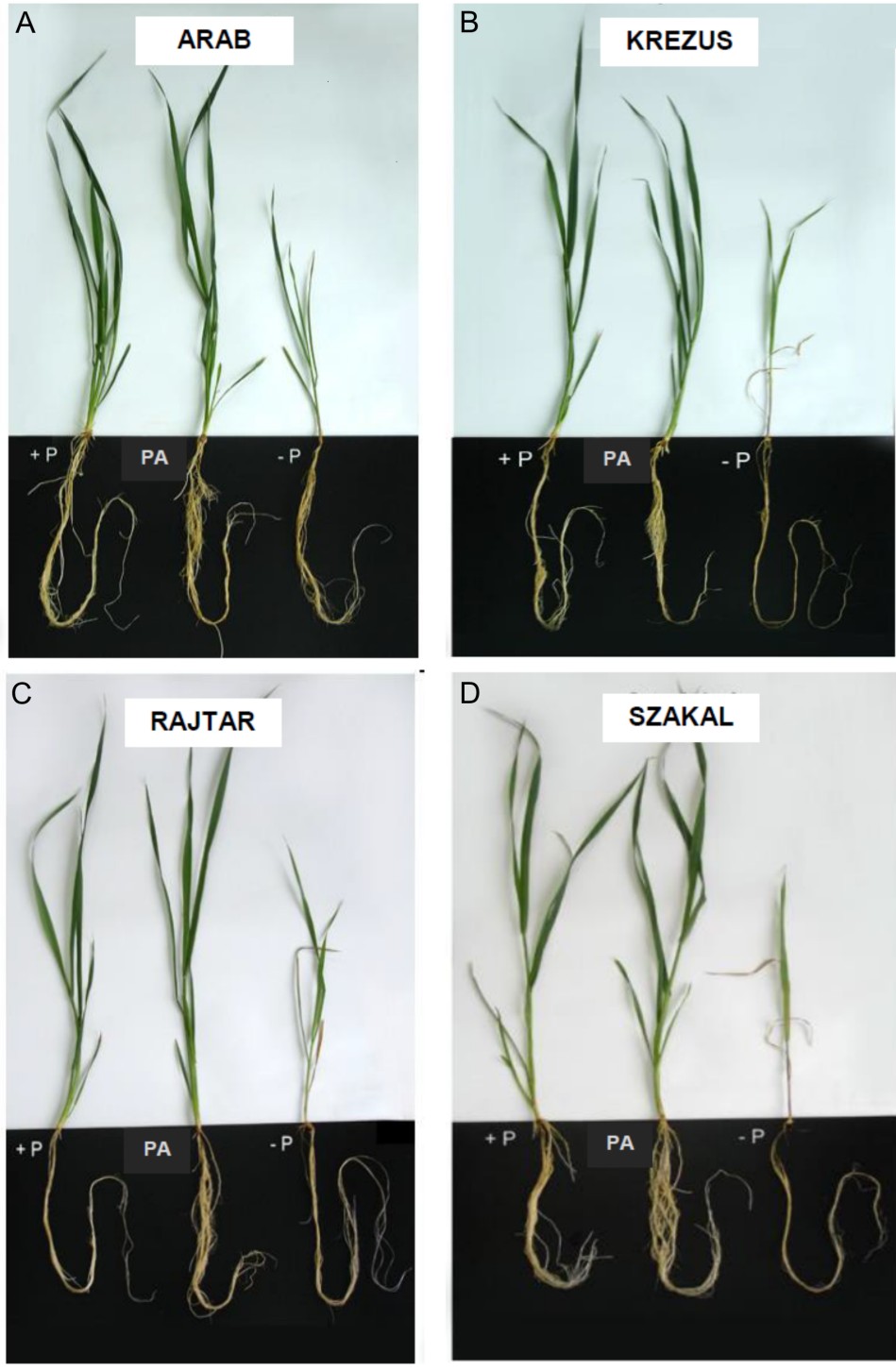

**Figure 2   Oat plants growth after 3 weeks of culture.** Oat cultivars (*Avena sativa* L., cv. Arab, Krezus, Rajtar and Szakal) grown for 3 weeks on a complete nutrient medium (+P), medium with phytic acid (PA) and without phosphate (−P).

**Table 1  Oat cultivars growth parameters.** Growth parameters of oat varieties (*Avena sativa* L., Arab, Krezus, Rajtar and Szakal) cultured for 3 weeks on complete nutrient medium (+P), medium with phytic acid (PA) and without phosphate (−P).

| Parameter | ARAB | | | KREZUS | | | RAJTAR | | | SZAKAL | | |
|---|---|---|---|---|---|---|---|---|---|---|---|---|
| | +P | PA | −P | +P | PA | −P | +P | PA | −P | +P | PA | −P |
| Shoot fresh mass (g) | 4.10 | 3.63 | 0.97* | 3.45 | 4.14 | 0.57* | 3.58 | 3.61 | 0.57* | 6.49 | 5.62 | 0.92* |
| Root fresh mass (g) | 2.08 | 1.84 | 1.39 | 2.27 | 2.43 | 1.00* | 1.68 | 2.22 | 0.81* | 2.87 | 2.47 | 0.75* |
| LAR (cm$^2$ A g$^{-1}$ W) | 19.2 | 19.1 | 16.4* | 20 | 16.3 | 12.7* | 21.6 | 22.6 | 25.2 | 17.4 | 19.4 | 19.9 |
| SWR (g WS g$^{-1}$W) | 0.7 | 0.7 | 0.4* | 0.5 | 0.6 | 0.3* | 0.7 | 0.6* | 0.5* | 0.6 | 0.6 | 0.5* |
| ULR (g$^{-1}$ΔdW cm$^2$A$^{-1}$ week$^{-1}$) | 0.003 | 0.003 | 0.005* | 0.003 | 0.004 | 0.001* | 0.003 | 0.003 | 0.003 | 0.004 | 0.003 | 0.004 |
| RGR (g ΔdW gW$^{-1}$ week$^{-1}$) | 0.06 | 0.05 | 0.08* | 0.06 | 0.06 | 0.02* | 0.07 | 0.07 | 0.07 | 0.07 | 0.06 | 0.06 |
| Root diameter (mm) | 1.3 | 1.2 | 0.9* | 1.2 | 1.3 | 0.7* | 1.1 | 1.2 | 0.7* | 1.4 | 1.3 | 0.7* |
| Root area (cm$^2$) | 16.2 | 14.9 | 15.7 | 18.6 | 18.8 | 13.9* | 15.2 | 18.9 | 12.3* | 20.0 | 18.9 | 11.4* |

**Notes.**
*Differences statistically important at 0.05.

in general similar to control, however, an increase in soluble sugars was observed for cv. Arab shoots and cv. Szakal roots (Figs. 3D–3E).

Organic acids/H$^+$ releases of oat roots were estimated using an agar sheet with pH indicator (bromocresol purple). There was no rhizosphere acidification, irrespectively of phosphate treatment, and even alkalization was observed (see Fig. S3).

Phosphate deficiency markedly increased acid phosphatase secretion and activity, which was visualized by an *in vivo* method (Fig. 4). The darkest red-brown colour indicating APase activity was observed in −P plant roots. APase tissue localization showed the highest activity in vascular tissues and in root epidermis of −P plants (Figs. 4N–4O).

Phosphate deficiency led to enhanced extracellular APase activity in the roots of all the studied cultivars (Figs. 5A–5C). APase activity was significantly higher already after one week of culture on −P medium in comparison to control (2.8 times higher for cv. Krezus and about two times higher for cv. Rajtar and Szakal). The highest activity was observed in cv. Krezus after 3 weeks of culture (7 times higher than +P). Plants grown on PA medium showed similar APase activity to control (Figs. 5A–5C). Extracellular APase activity exhibited a high negative correlation with tissue Pi content. Kinetic assay of extracellular APase activity showed at least two times higher Vmax value for −P plants of all the tested plant cultivars, as compared to control (and even three times higher for cv. Krezus and Szakal) (see Fig. S4). The Michaelis constant ($K_m$) was markedly lower only for cv. Krezus and Rajtar −P plants. PA plants had similar $V_{max}$ and $K_m$ values compared to control plants (see Fig. S4).

Insufficient phosphate supply did not cause such a significant increase in intracellular phosphatase activity (Figs. 5D–5I) like extracellular enzymes. APase activity in −P plant shoots was increased only for cv. Krezus after one week of culture (1.6 times higher than in control) (Figs. 5D–5F). After 2 weeks of −P culture, a higher increase of APase activity was observed in cv. Rajtar (two times higher than in +P) and cv. Szakal (3.5 higher) shoots. After 3 weeks of −P culture, an increase in enzyme activity was observed only for shoots of

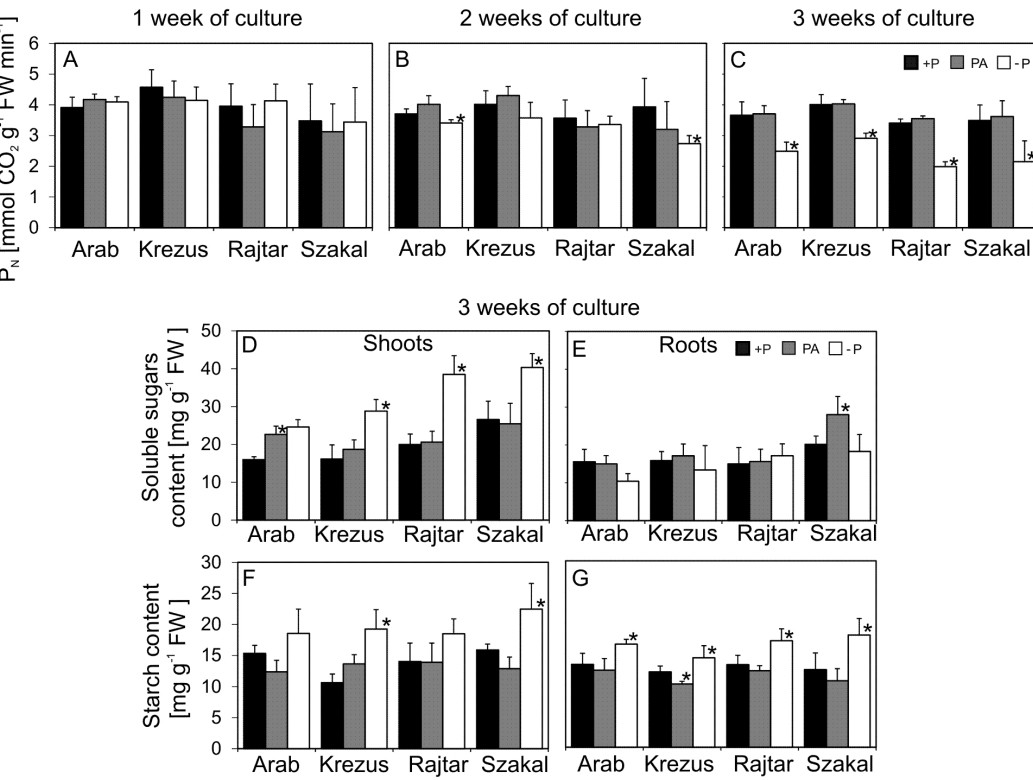

**Figure 3 Photosynthetic activity (PN) (A–C), soluble sugars (D–E) and starch (F–G) content in shoots and roots.** Oat cultivars (*Avena sativa* L., cv. Arab, Krezus, Rajtar and Szakal) grown in a complete nutrient medium (+P), medium with phytic acid (PA) and without phosphate (−P) (means ± SD). *Differences statistically significant at 0.05.

cv. Krezus, whereas a decrease in the shoots was recorded in cv. Rajtar and Szakal (down to 50% of control). APase activity in the shoots of PA plants was similar to control during two weeks of culture and decreased only after 3 weeks in cv. Rajtar and Szakal (78% and 52% of control, respectively) (Figs. 5D–5F).

APase activity in the roots of plants grown on −P medium for 1–2 weeks was similar to control for all the studied oat cultivars, except cv. Krezus, where the activity was 1.5–2 times higher than in control plants and cv. Rajtar where the activity was reduced down to 44% in comparison to +P plants (Figs. 5G–5I). The highest increase in this enzyme activity was observed in cv. Szakal after 3 weeks of −P culture (six times higher than in control), whereas only a 1.7 times increase was observed for cv. Krezus and Rajtar. Plants grown on medium with phytate showed similar activity of root APase compared to control (Figs. 5G–5I).

Three main APase isoforms (about 95 kDa, 70 kDa and 27 kDa) were detected in native electrophoresis gels, independently of phosphorus treatment, however, the activity of the smallest one was higher in roots than in shoots, especially in −P and PA plants (except cv. Szakal) (Fig. 6).

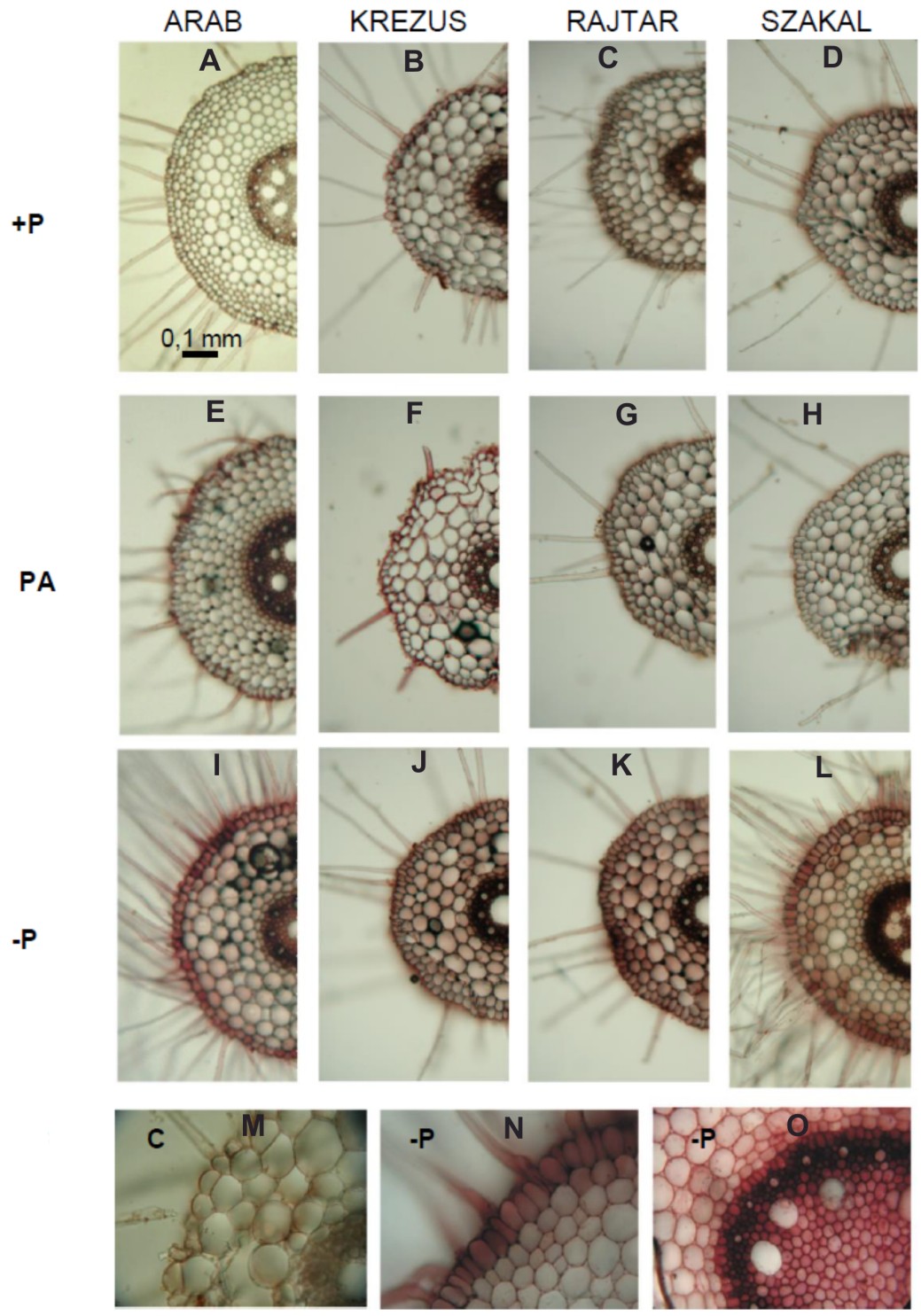

**Figure 4** *In vivo* **staining for acid phosphatase activity in root cross sections.** Oat cultivars (*Avena sativa* L., cv. Arab, Krezus, Rajtar and Szakal) cultured for 1 week on a complete nutrient medium (+P) (A–D), medium with phytic acid (PA) (E–H) and without phosphate (−P) (I–L, N–O). The dark red-brown colour indicates acid phosphatase activity in the root tissues, as compared to the heat-killed tissue–control (M).

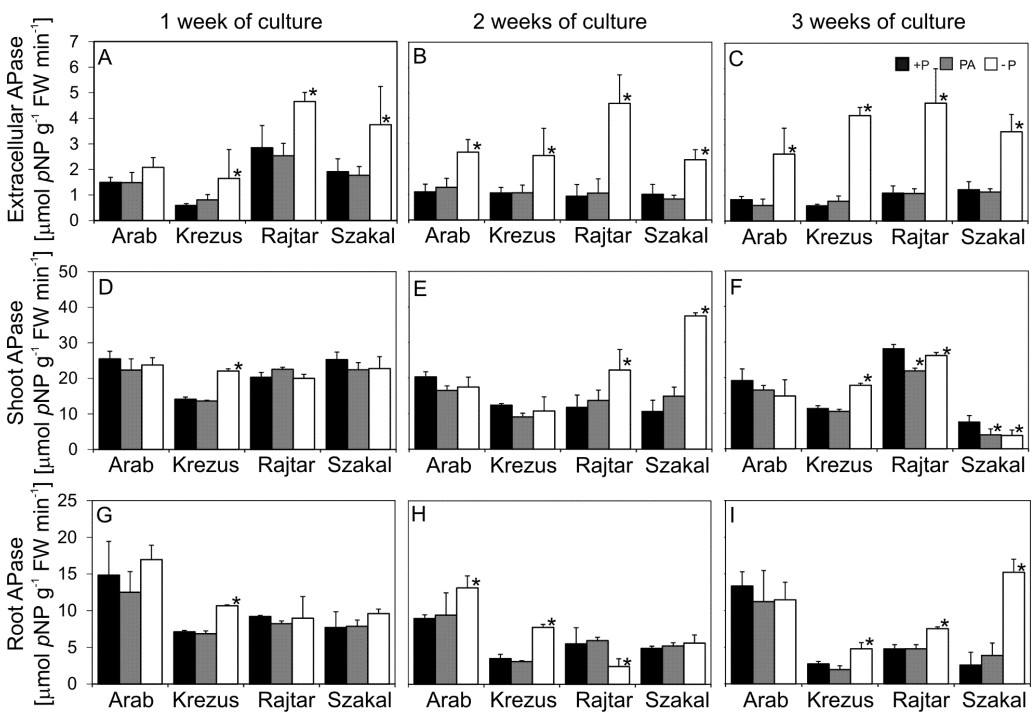

**Figure 5** Extracellular (A–C) and intracellular acid phosphatase activity in shoots (D–F) and roots (G–I). Oat cultivars (*Avena sativa* L., cv. Arab, Krezus, Rajtar and Szakal) grown for 1, 2 and 3 weeks on a complete nutrient medium (+P), nutrient medium with phytic acid (PA) or without phosphate (−P) (means ± SD). *Differences statistically significant at 0.05.

The effect of mycorrhizal colonization (M) on growth parameters and Pi content in two selected oat cultivars (Arab and Krezus) was also investigated. Shoot height of both oat cv. was neither affected by phosphate nutrition nor by AM fungus (Table 2). The longest roots were observed in PA and PA$_M$ plants of both oat cultivars (about 1.4 times longer than in control). Shoot fresh and dry weight was two times lower in −P (both cultivars) and −P$_M$ plants (only cv. Arab) (Table 2). Mycorrhization increased shoot mass 1.7 times in comparison to non-infected plants grown on phytate medium and without phosphate only in cv. Krezus. Root fresh weight was 1.4 times higher in −P plants (cv. Arab) and 2.5 higher in PA$_M$ plants (cv. Krezus). Mycorrhization increased root mass only in PA$_M$ plants (1.6 higher than in PA plants) in cv. Krezus (Table 2).

Phosphate deficiency decreased Pi content in shoots, irrespective of mycorrhizal colonization (Fig. 7A). A higher decrease (down to 11–14% of control) was observed in cv. Arab shoots than in Krezus (down to 21–24%). PA plants also showed a decreased Pi content in the shoots (44% and 29% for cv. Arab and Krezus, respectively). Mycorrhizal infection led to a decrease in Pi content (down to 68% of control) in the shoots of cv. Krezus plants grown on phytic acid when compared to +P plants. However, mycorrhization resulted in a higher Pi content in PA$_M$ plants (1.8 times higher and 2.4 higher for cv. Arab and Krezus, respectively) in comparison to non-infected plants (PA) (Figs. 7A–7D). Pi content in oat roots grown on phytic acid and without phosphate decreased (irrespective

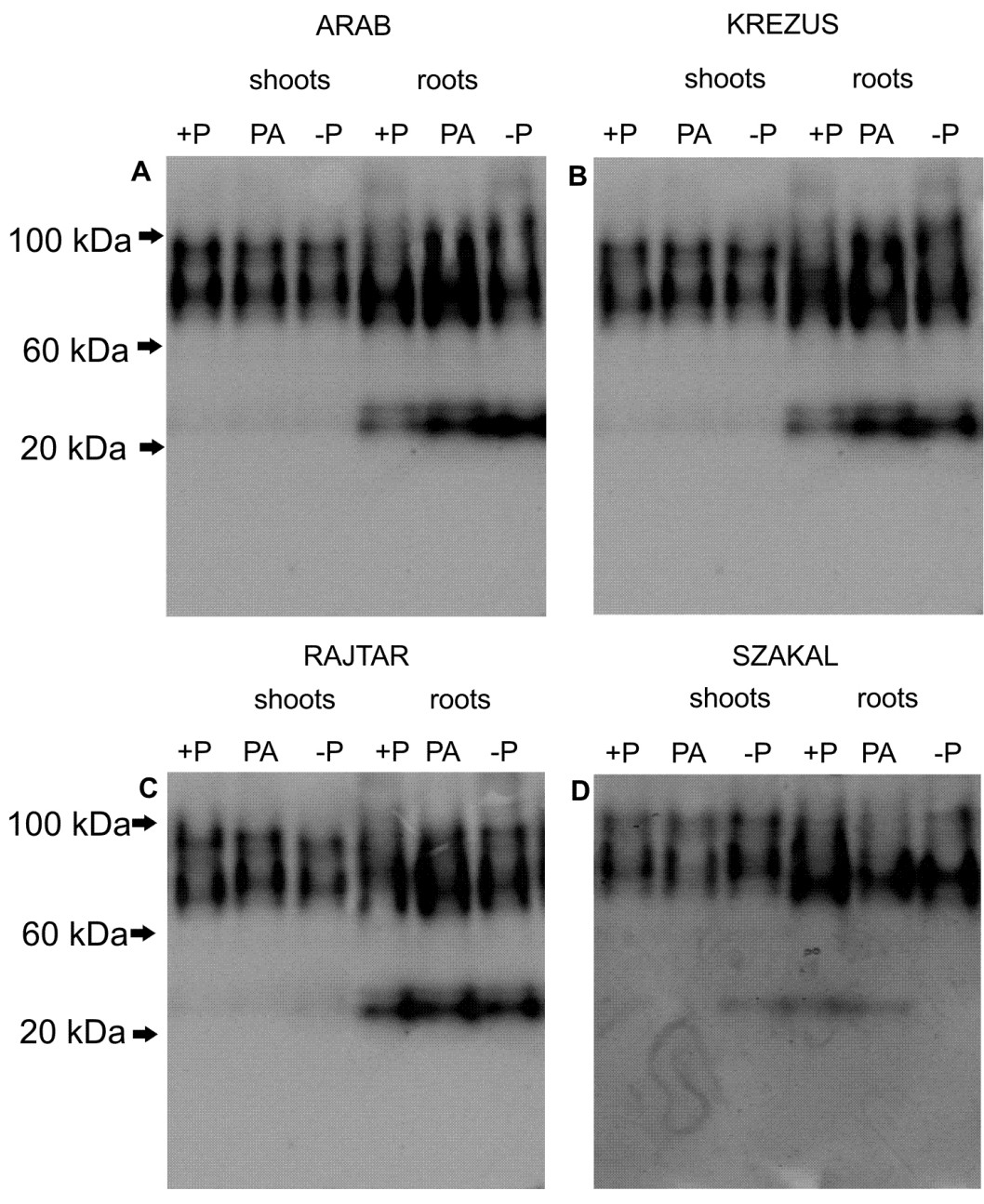

**Figure 6  Profile of APase isoforms in crude protein extracts from shoots and roots.** Oat cultivars (*Avena sativa* L., cv. Arab, Krezus, Rajtar and Szakal) cultured for 3 weeks on a complete nutrient medium (+P), medium with phytate (PA) and without phosphate (−P). Protein extracts from shoots (10 μg protein per lane) and roots (6 μg protein per lane) were run on native discontinuous PAGE and stained for APase activity using 4-methylumbelliferyl phosphate and visualized under UV light.

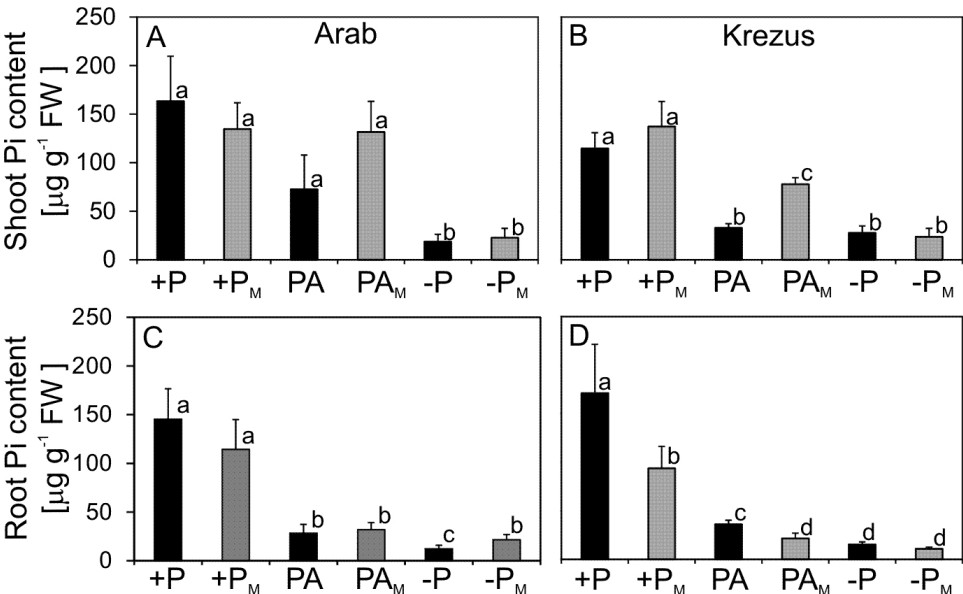

**Figure 7 Inorganic phosphate (Pi) content in shoots (A–B) and roots (C–D) of plants treated (M) or not treated with AM fungi.** Oat cultivars (*Avena sativa* L., cv. Arab and Krezus) grown for 5 weeks on a sand complete nutrient medium (+P), medium with phytic acid (PA) and without phosphate (−P) non-inoculated or inoculated (M) with Glomus intraradices. Significant differences are indicated by different letters.

of mycorrhiza), however, Pi content was lower in −P plants than in PA plants. The positive effect of mycorrhization was observed only in −P$_M$ plants of cv. Arab (1.8 times higher Pi content in comparison to −P plants) (Fig. 7C).

## DISCUSSION

Understanding plant responses to low phosphate nutrition has been the subject of considerable research efforts for many decades. Recently, molecular mechanisms of model plant (mainly *A. thaliana*, but also *O. sativa*) responses to Pi deficiency have been intensively studied, whereas acclimation mechanisms of crop plants, like oat (*A. sativa*) to low-Pi stress remain poor known. Oat plants are usually grown on low-productive, poor soils with Pi shortage (*Butt et al., 2008*), however, their responses to Pi starvation are rarely investigated. The present study contributes to the understanding of oat acclimation mechanisms to low Pi nutrition. Our results indicated that acid phosphatases play the most important role in oat acclimation to Pi deficiency, mainly extracellular enzymes involved in Pi acquisition. The study showed that four different oat cultivars could effectively utilized phytates, organic P forms. Phytates are generally not an easily available P form for other cultivated crop plants (*George et al., 2008*; *Richardson et al., 2009*). Studied oat cultivars developed similar acclimation mechanisms to Pi deficiency, however, there were differences in the usage of shoot/root or internal/extracellular APases pools.

Pi deficiency in nutrient media caused severe Pi content decrease in plant tissues (down to 4% of control after three weeks of culture). Surprisingly, all the studied oat cultivars

**Table 2** Growth parameters of oat plants (*Avena sativa* L. cv. Arab, Krezus) inoculated (*M*) or not inoculated with *Glomus intraradices*.

| Parameter | +P | +P$_M$ | PA | PA$_M$ | −P | −P$_M$ |
|---|---|---|---|---|---|---|
| | | | **ARAB** | | | |
| Shoot height (cm) | 62.0[a] | 64.8[a] | 59.4[a] | 65.2[a] | 65.1[a] | 62.1[a] |
| Root length (cm) | 14.3[a] | 17.6[a] | 22.8[b] | 23.2[b] | 19.5[a] | 18.1[a] |
| Shoot fresh mass (g) | 8.00[a] | 8.24[a] | 8.71[a] | 8.13[a] | 4.26[b] | 4.6[b] |
| Root fresh mass (g) | 2.67[a] | 1.8[a] | 1.56[a] | 1.97[a] | 3.82[b] | 2.69[a] |
| | | | **KREZUS** | | | |
| Shoot height (cm) | 64.1[a] | 63.9[a] | 69.8[a] | 67.9[a] | 60.9[a] | 62.5[a] |
| Root length (cm) | 16.9[a] | 17.8[a] | 20.5[b] | 23.2[b] | 17.9[a] | 17.9[a] |
| Shoot fresh mass (g) | 8.1[a] | 7[a] | 10[a] | 18.2[b] | 4.7[c] | 7.7[a] |
| Root fresh mass (g) | 2.34[a] | 1.88[a] | 3.72[a] | 5.82[b] | 1.78[a] | 2.57[ac] |

**Notes.**

Oat varieties (Arab and Krezus) grown for 5 weeks on sand watered with complete nutrient medium (+P), medium with phytic acid (PA) or medium without phosphate (−P) not inoculated or inoculated ($_M$) with *Glomus intraradices*. Different letters indicate significant differences ($P < 0.05$).

(Arab, Krezus, Rajtar, Szakal) grown with phytic acid showed similar P content (which only slightly decreased after three weeks of culture) to the plants supplied with easily accessible inorganic P (Fig. 1, Fig. S1). These results suggest that such organic P is a potentially available source of P for oat plants. Similar effects were also observed for barley plants in our previous study (*Ciereszko, Żebrowska & Ruminowicz, 2011*) and in another study on rapeseed (*Zhang et al., 2010*). However, the ability of utilizing organic P is not very common in plants. For example, Pi content in shoots of wheat plants grown on organic source of P was about three times lower as compared to control in the study of *George et al. (2008)*, whereas in our study, some PA plants showed even higher Pt content (see Fig. S1).

Pi acquisition by plants may be enhanced due to root morphological/architectural adaptations that increase the root surface area. The studied oat cultivars grown on −P medium showed significantly decreased shoot weight and height (Fig. 2), whereas root length was hardly affected by Pi deficiency and even root elongation was observed for *A. sativa* cv. Arab (associated with smaller root diameter); the growth of PA plants was similar to control (Table 1). The ability to maintain, or even extend, root growth is one of the important strategies to improve Pi uptake efficiency. Our results showed a pronounced increase in root/shoot biomass (and length) ratio for the four studied oat cultivars under low Pi supply (see Fig. S2), which corresponded with low P concentration in the tissues. However, root architectural response may vary between plant species and even genotypes, as indicated in other reports (*Péret et al., 2014* and articles cited therein). Some plants tend to reduce primary root growth and extend lateral root and root hair (e.g., *A. thaliana*), whereas other increase both primary and lateral root growth (e.g., *O. sativa*) (*Chevalier et al., 2003*; *Dai et al., 2012*; *Péret et al., 2014*; *Hoehenwarter et al., 2016*) or increase primary root growth and reduce lateral root formation (e.g., in maize, *Li et al., 2012*). Among the studied oat cultivars, Pi deficiency had the lowest impact on the growth of oat cv. Arab (Table 1). It has been reported that plants growing in Pi-deficient conditions often allocate more assimilates to root growth and tend to have fine roots (of small diameter),

and therefore a large surface area. However, in our experimental conditions we did not observe larger root surface areas (except for cv. Krezus). Moreover, calculated areas were even lower for oat cv. Rajtar and Szakal (Table 1, Table S1). We have previously investigated the growth of bean, barley, cucumber, oat and wheat plants under Pi-deficient conditions, where the reduction of shoot mass was usually accompanied by root elongation stimulation and a significant increase of the root/shoot weight ratio (*Ciereszko, Miłosek & Rychter, 1999*; *Ciereszko, Janonis & Kociakowska, 2002*; *Ciereszko, Szczygła & Żebrowska, 2011*; *Ciereszko, Żebrowska & Ruminowicz, 2011*; *Żebrowska, Bujnowska & Ciereszko, 2012*). However, in spite of the currently available data, genetic control of primary and lateral root development in monocot/cereal species under Pi deficiency remains poorly understood (*Li et al., 2012*; *Péret et al., 2014*). Recent quantitative trait loci analyses in wheat or other cereals have shown that Pi utilization efficiency is a complex, polygenic trait (*Aziz et al., 2014*).

Pi deficiency markedly decreased growth rate parameters, especially for cv. Krezus and to a lesser extend for cv. Rajtar and Szakal. In contrast, unit leaf rate and relative growth rate of cv. Arab was even higher, as compared to control (Table 1) (see also Table S1). However, culture on PA media not affected growth parameters of studied oat plants. High RGR values are usually observed in plants grown under optimal conditions (*Shipley & Keddy, 1988*). Therefore, an increase of this parameter observed in −P oat plants (cv. Arab) could indicate that these plants are less susceptible to Pi deficiency. Similar results were obtained for potato genotypes grown under Pi deficiency, where high RGR values were associated with higher P-utilization efficiency (*Balemi & Schenk, 2009*). Furthermore, Pi-deficient legumes with high RGR rates (*Lotus australis*) showed both morphological and physiological (i.e., root carboxylate dynamics) root adaptations, whereas plants with lower RGR (i.e., *Cullen cinereum*) demonstrated only morphological root adaptations (*Suriyagoda et al., 2012*). Recent studies suggested that differences in P-use efficiency between two contrasting wheat genotypes resulted from different Pi and organic P allocation patterns. Shoot biomass production in both cultivars was surprisingly similar, irrespectively of P supply, whereas Pi efficient genotype showed higher root biomass (*Aziz et al., 2014*).

The decrease in biomass and growth rate indicators under Pi starvation could be caused by limited PAR absorption (due to the smaller leaf area and their number) or less efficient use of absorbed radiation; therefore, photosynthetic intensity was measured. A significant decline (down to 60% of control) of $P_N$ occurred in leaves of all studied oat cultivars after 3 weeks of growth on Pi deficient medium (Fig. 3), however, photosynthetic pigments concentration was not affected (see Table S2). In contrast, $P_N$ of PA plants was similar to control, which indicated that Pi acquired from phytate was effectively used for oat plants metabolism. Early and intermediate Pi deficiency usually has no influence on photosynthetic $CO_2$ exchange, or may lead to increased photosynthetic intensity, however, severe Pi deficiency leads to a decline in $P_N$, as observed in previous studies (*Ciereszko et al., 1996*; *Maleszewski et al., 2004*; *Li et al., 2006*). Interestingly, high photosynthetic activity under Pi deficiency could be obtained as a result of phospholipid replacement with galactolipids and sulfolipids during leaf development, as observed in *Proteaceae* species (*Lambers et al., 2012*); similar replacements of lipid components in plasma membranes were also reported for oat plants (*Andersson et al., 2005*). Changes in photosynthetic

activity in Pi-deficient plants often cause redistribution of assimilates. The accumulation of soluble sugars and starch in leaves and starch in roots was demonstrated in −P oat cultivars (Figs. 3D–3G). The observed differences in sugar contents could be both the result of sugar metabolism modification and/or changes in sugar distribution between plant organs. High sugar content in roots of Pi-deficient plants may be caused by increased assimilate transport to these organs, and enhanced activity of sucrose hydrolysing enzymes, as previously observed for bean (*Ciereszko et al., 1996*; *Ciereszko, Miłosek & Rychter, 1999*; *Ciereszko & Barbachowska, 2000*). Pi depletion changed the expression of many genes involved in biosynthesis and degradation of starch and sucrose, and caused starch and sugar accumulation in *Arabidopsis* and rice leaves (*Ciereszko & Kleczkowski, 2005*; *Misson et al., 2005*; *Hammond & White, 2008*; *Park et al., 2012*). Moreover, sucrose transport modification in the phloem, observed under Pi deficiency, might also initiate sugar signalling cascades that alter multiple gene expression involved in plant response to Pi deficiency (*Hammond & White, 2008*; *Park et al., 2012*).

Pi use efficiency is possible due to Pi remobilization from various cell compartments (mainly vacuole) and from older to younger plant organs by internal acid phosphatases (*Duff, Sarath & Plaxton, 1994*; *Tian & Liao, 2015*). Pi deficiency affected the intracellular APase activity in roots and shoots of oat plants, however, these changes were not as significant as in case of extracellular APases (Fig. 5). Generally, higher APase activity in shoots was observed only for oat cv. Krezus and in roots of three oat cultivars (except cv. Arab) (Figs. 5D–5I). APase enzyme activities in the plants grown on PA nutrient media were similar to control values. Three APase isoforms were found in shoot and root extracts of all the studied oat cultivars. The smallest isoform was strongly expressed in roots, generally irrespectively of P treatment (Fig. 6). No significant differences in isoform patterns between oat cultivars under study were observed, similar to our previous results (*Ciereszko, Żebrowska & Ruminowicz, 2011*; *Żebrowska, Bujnowska & Ciereszko, 2012*).

The increased Pi uptake capacity of roots is dependent not only on the increased nutrient absorption surface, but also on the metabolic capacity to release a wide range of organic anions, protons (or enzymes) to the ground, which play a key role in increasing the mobilization of P from sparingly soluble sources in soil (*Wang et al., 2015*; *Wang et al., 2017*; *Tian & Liao, 2015* and articles cited therein). However, our measurements did not show a distinct effect of the insufficient Pi feeding on rhizosphere acidification (due to organic acid or $H^+$ release) (see Fig. S3), which indicated that oat cultivars rather did not respond to Pi starvation by increased protons or organic acid release from the root system (rhizoplane), at least in those experimental conditions. Recent studies conducted on different crops, including oat, showed that organic anion exudation to the rhizosphere played a minor role in improving Pi availability and uptake in agricultural soils (*Wang et al., 2015*; *Wang et al., 2016*). The latter authors indicated that other factors, such as root morphology, pH (associated with $H^+$ release by the roots) or acid phosphatases might play a key role in the effective utilization of phosphorus (*Wang et al., 2015*; *Wang et al., 2016*).

Mycorrhizal symbiosis enhances nutrient acquisition in host plants by exploring a larger soil volume by extending fungi hyphae beyond P-depletion zone in the soil (*Smith & Smith, 2011*; *Yang et al., 2012*; *Ceasar et al., 2014*). As an example, foxtail millet plants

inoculated with *Glomus mosseae* showed about 30% increase in seed yield when compared to non-infected plants (*Ceasar et al., 2014*). In our study, no significant positive effect of mycorrhization on plant growth was observed - only cv. Krezus plants grown on phytic acid showed almost two times higher shoot mass when compared to non-infected plants (Table 2). Consistent with that result, shoot Pi content in this cultivar was also increased (Fig. 7), which indicated that colonization by AM fungi increased plant acquisition of organic P forms. This was consistent with previous results, also obtained in oat plants, especially after dual inoculation with *Glomus etunicatum* and *Glomus intraradices* (*Joner, Van Aaerle & Vosatka, 2000*; *Khan, Ahmad & Ayub, 2003*). Phytate utilization was also enhanced in trifoliate orange (*Poncirus trifoliata*) inoculated with AM fungi (due to the increased phytase and APase activity in the roots and substrate (*Shu, Wang & Xia, 2014*). Interestingly, some studies showed a negative influence of mycorrhization on host plant growth, as a result of sink competition for photosynthates or pairing of host and fungal symbiont (*Rai, Rai & Rakshit, 2013*; *Nouri et al., 2014* and articles cited therein). In contrast, wheat plants inoculated with AM fungi had higher biomass and grain yield, regardless of P application (*Hu et al., 2010*).

Enzyme secretion by plant roots (or *rhizosphere* microorganisms) may also increase Pi uptake from organic phosphorus sources in the soil. The increase of APases activity and secretion seems to be a common reaction to Pi deficiency and was reported in many works, also in our previous studies conducted on barley and wheat (*Ciereszko, Szczygła & Żebrowska, 2011*; *Ciereszko, Żebrowska & Ruminowicz, 2011*). The increase in extracellular APase activity was already observed after one week of growth in −P medium (Fig. 5), which indicated that enzyme secretion from the roots is an early plant response to Pi deficiency. High APase activity in the root epidermis of phosphate-depleted plants observed in root transverse sections (Fig. 4) indicated that these enzymes could be secreted by oat roots to the ground. Our previous study conducted on whole roots of −P oat showed high APase activity not only in roots, but also within a few millimetres away from plant roots (*Żebrowska, Bujnowska & Ciereszko, 2012*), suggesting an increased production and secretion of enzymes by these roots under Pi deficiency. White lupine roots also showed the highest APase activity in root epidermis and root hairs of both proteoid and non-proteoid roots (*Wasaki et al., 2009*; *Tang et al., 2013*). It seems that the production of APases associated with the root surface (rather than those released to the rhizosphere) is a better strategy to efficiently acquire Pi liberated from organic P. Recent studies have indicated that purple APases secreted by *A. thaliana* under Pi deficiency remains associated with the surface of root epidermal cells (AtPAP10) or cell wall–targeted (AtPAP25) and play an important role in plant acclimation to Pi deficiency (*Wang et al., 2011*; *Del Vecchio et al., 2014*).

The highest extracellular acid phosphatase activity was observed after 3-week culture of oat cv. Krezus, when compared to other cultivars (Figs. 5A–5C). Oat genotypes with high efficiency of extracellular APase activity or enzyme secretion from the roots to the soil could probably significantly enhance the availability of phosphorus from organic fertilizers. Our study demonstrated that all oat cultivars can use phytate as the sole source of phosphorus and can grow and develop well in such conditions, in contrast to

other plants (*George et al., 2008*). The highest extracellular APase activity was observed in younger, growing parts of the roots. *Tang et al. (2013)* also showed the induction of APase activity in the elongation zone of the root tips and root meristems of lupin. A significant increase in APase production and secretion under Pi deficiency is well documented in numerous plants (*Żebrowska & Ciereszko, 2009*; *Tran, Hurley & Plaxton, 2010* and articles cited therein). It was shown that the overexpression of gene(s) encoding APase in plant roots (including root hairs) increased Pi efficiency, when compared to control (*Wasaki et al., 2009*; *Zhang, Liao & Lucas, 2014*; *Lu et al., 2016*). Interestingly, the high APase and phytase activity along with fine root morphology of *Polygonum hydropiper* (mining ecotype) are responsible for organic phosphorus assimilation capacity and, as suggested, may be used in phytoremediation of areas polluted with organic P (*Ye et al., 2015*). However, some authors reported no relationship or a negative relationship between acid phosphatase activity and Pi use efficiency, e.g., clover genotypes with contrasting Pi-uptake efficiency did not differ in APase activity (*Hunter & McManus, 1999*; *Yan et al., 2001*). What is more, there is no evidence that the expression of up-regulated phosphatase genes under Pi deficiency is higher in P-efficient rice genotypes (*Rose et al., 2013*). Additionally, the differences in APase activities observed in laboratory conditions are not that evident when plants are grown in the soil (*George et al., 2008*). Therefore, the detailed function of APases in the acclimation of various crop plants to Pi deficiency is still not well understood and is under intensive investigations. A more detailed analysis of APases/phytases secreted by oat roots is necessary. In a recent study, the function of a novel secreted rice purple APase, OsPAP10c, was investigated in the utilization of external organic P (*Lu et al., 2016*). It was demonstrated that the overexpression of *OsPAP10c* significantly increased APase activity in rice tissues, mainly on the root surface, but also in culture media. Other recent studies indicated that the concentration of rhizosphere APases was positively correlated with plant-available phosphorus fractions and Pi absorption (including *Brassica napus* and *A. sativa*) in Pi-deficient soils (*Wang et al., 2016*).

## CONCLUSION

In conclusion, the studied oat cultivars grew well both on the medium containing inorganic P as well as organic P (phytate), and only slightly differed in terms of acclimation to moderate Pi deficiency. Pi sources are non-renewable and organic P compounds are usually in excess. Therefore, it is currently important to investigate and select cereal plant genotypes tolerant to Pi depletion in the soil, which are able to develop in a phytate-rich ground in order to sustain the yield of common crops. The study provided useful information for future investigation of oat behaviour under field conditions. Generally, small variations in secretion, localization and activity of APases were observed between oat cultivars; however, in certain conditions they used different pools of acid phosphatases to acquire Pi from external or internal P sources. The most important component of the acclimation mechanism of oat to low Pi conditions was the enhanced activity of mainly extracellular acid phosphatases.

**Abbreviations**

| | |
|---|---|
| **APase** | Acid phosphatase |
| **LAR** | Leaf area ratio |
| **P** | Phosphorus |
| **PA** | Plants cultured on medium with phytate |
| **PAGE** | Polyacrylamide gel electrophoresis |
| **PA$_M$** | Oat plants cultured on medium with phytate inoculated with *Glomus intraradices* |
| **Pi** | Inorganic phosphate |
| **P$_N$** | Net photosynthesis intensity |
| **Pt** | Total phosphorus |
| **+P** | Phosphate-sufficient plants (control) |
| **+P$_M$** | Phosphate-sufficient oat plants inoculated with *Glomus intraradices* |
| **−P** | Phosphate-deficient plants |
| **−P$_M$** | Phosphate-deficient oat plants inoculated with *Glomus intraradices* |
| **RGR** | Relative growth rate |
| **SWR** | Stem weight ratio |
| **ULR** | Unit leaf rate |

## ACKNOWLEDGEMENTS

We wish to thank the cereal seeds producers IHAR (Strzelce, Poland) and DANKO (Choryn, Poland) for oat seeds used in the experiments. We thank for technical assistance by Magdalena Klimiuk, Beata Kuciejczyk (growth and productivity parameters measurements), and Martyna Kempista, Paula Trebicka (photosynthesis and sugar estimations).

### Funding

This work was supported by the Grant DEC-2012/07/N/NZ9/00972 from the National Science Center (NCN, to Ewa Żebrowska), Poland. The funders had no role in study design, data collection and analysis, decision to publish, or preparation of the manuscript.

### Grant Disclosures

The following grant information was disclosed by the authors:
National Science Center: DEC-2012/07/N/NZ9/00972.

### Competing Interests

The authors declare there are no competing interests. There are no competing interests among seed producers and the authors of the manuscript (and the Institute of Biology UwB, Poland)—all oat seeds were provided as a gift from seeds producers: IHAR (Strzelce, Poland) and DANKO (Choryn, Poland).
## Author Contributions

- Ewa Żebrowska conceived and designed the experiments, performed the experiments, analyzed the data, contributed reagents/materials/analysis tools, wrote the paper, prepared figures and/or tables.
- Marta Milewska performed the experiments.
- Iwona Ciereszko conceived and designed the experiments, analyzed the data, contributed reagents/materials/analysis tools, wrote the paper, prepared figures and/or tables, reviewed drafts of the paper.

## Data Availability

The raw data has been provided as a Supplemental File.

## Supplemental Information

Supplemental information for this article can be found online at http://dx.doi.org/10.7717/peerj.3989#supplemental-information.

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
