# Peer review of "Mechanisms of oat (Avena sativa L.) acclimation to phosphate deficiency"

_PeerJ, doi:10.7717/peerj.3989_

## Round 0.1 · original submission · Minor Revisions

Please answer the questions of the referees listing the responses one by one, and take care of the mistakes and omissions.

·

Basic reporting

The manuscript reports the mechanisms of oat acclimation to phosphate deficiency. The major findings include the capacity of oats to use organic phosphate (phytate) as a source of phosphorus and the role of different pools of acid phosphatases in acquiring phosphorus from soil. The paper is prepared professionally, minor language editing is recommended (e.g. "another enzymes" in the abstract, line 57, etc.). The statement that acclimation of cereals to phosphate deficiency remains undiscovered sounds a bit too strong. There are many publications in this field, and the authors cite the most important papers in their manuscript. The manuscript contains valuable sets of data that can be useful for agronomists and plant physiologists involved in the problem of phosphate deficiency in cereals and in oat in particular.

Experimental design

The experimental design is acceptable. Four cultivars have been used. The parameters studied mostly refer to the general classical physiology. The biochemical characterization of isoforms of acid phosphatases adds an important information to this study. The methods are described with sufficient detail. This research fills some existing knowledge gaps in phosphorus acquisition in cereal plants.

Validity of the findings

The major findings in this study are valuable. They include the capacity of oats to use organic phosphate (phytate) as a source of phosphorus and the role of different pools of acid phosphatases in acquiring phosphorus from soil.

Additional comments

In general, this manuscript contains important sets of data on phosphorus acquisition by oat plants. The study has a value for practical applications. It is useful for the agronomists that face the problems of phosphate deficiency in their field practice.

Reviewer 2 ·

Basic reporting

The topic of the article is interesting and it is suitable for publishing in Peer J.
The presented studies concern on growth parameters and physiological adaptations which enable four commercial cultivars of oat to grow under Pi deficiency. The authors postulated that extracellular phosphatases play significant role in acclimation of oat plants to low Pi regime.
Although the method of acid phosphatases assays is quite old-fashion, the concept and results are valuable and might be good base for application in agriculture. A better understanding of the mechanisms of plants responses to Pi deficiency is essential to develop strategies to increase crop yield. It would be worth to carry out some genetics approaches to elucidate mechanism of oat responses to Pi starvation at molecular level (i.e. identification of AP sequences in oat’s genome, analyses of AP genes expression in different Pi nutrition, Western Blot to determine AP proteins in more quantitative way).
The introduction is written well and purpose of the study is clearly presented. Also results were analyzed properly with adequate statistical methods. The discussion was based on updated literature and supports the research hypothesis.

Experimental design

No comment

Validity of the findings

No comment

Additional comments

Minor changes have been suggested:
Line 198-200: Please explain why you used different concentrations of inorganic (1 mM KH2PO4) and organic (0,1 mM phytic acid) as a P source? Is 1 mM concentration of Pi enough optimal for crops?
Line 425: “…oat cultivars could effectively utilized phytates, organic P forms generally not easily available to other plants”. This sentence should be rewritten. It seems that only oat might use phytates as a P source? Are there any specific phytases from oat that mobilize organic P reserves in soil? A reference is needed.
Line 613: Abbrevations:
PN – is missing
PM – is missing
PAM- is missing

---

## Round 0.2 · Minor Revisions

Although the concerns and suggestions made by the referees have been mostly taken care of, there are still a few corrections needed as well as a careful check of the citation style used in the text required:

For example: line 257: Km and Vmax should be with subscripts
line 270: naphthyl
line 613: In future studies, a more detailed analysis.....is required.

I have appended a tracked changes document of my edits (converted to a PDF) and staff will send you the original Word doc. Please also note the question on line 580.

---

## Round 0.3 · accepted · Accept

The remaining few errors should be corrected during production:

line 43; In addition,.....
line 580: but also within a few millimetres away from plant roots (Żebrowska et al. 2012), suggesting an increased production and secretion...